# Molecular Changes in Retinoblastoma beyond *RB1*: Findings from Next-Generation Sequencing

**DOI:** 10.3390/cancers13010149

**Published:** 2021-01-05

**Authors:** Jasmine H. Francis, Allison L. Richards, Diana L. Mandelker, Michael F. Berger, Michael F. Walsh, Ira J. Dunkel, Mark T. A. Donoghue, David H. Abramson

**Affiliations:** 1Ophthalmic Oncology Service, Department of Surgery, Memorial Sloan Kettering Cancer Center, New York, NY 10065, USA; abramsod@mskcc.org; 2Department of Ophthalmology, Weill Cornell Medical Center, New York, NY 10065, USA; 3Marie-Josee and Henry R. Kravis Center for Molecular Oncology, Memorial Sloan Kettering Cancer Center, New York, NY 10065, USA; richara4@mskcc.org (A.L.R.); donoghum@mskcc.org (M.T.A.D.); 4Department of Pathology, Memorial Sloan Kettering Cancer Center, New York, NY 10065, USA; mandelkd@mskcc.org (D.L.M.); bergerm1@mskcc.org (M.F.B.); 5Department of Medicine, Memorial Sloan Kettering Cancer Center, New York, NY 10065, USA; walshm2@mskcc.org; 6Department of Pediatrics, Memorial Sloan Kettering Cancer Center, New York, NY 10065, USA; dunkeli@mskcc.org; 7Department of Pediatrics, Weill Cornell Medical Center, New York, NY 10065, USA

**Keywords:** retinoblastoma, vitreous seeds, BCOR, next-generation sequencing, copy number variations

## Abstract

**Simple Summary:**

The gene causing retinoblastoma was the first tumor suppressor cloned (1986) and because retinoblastoma is the classic example of autosomal dominant inheritance, there has been little research on non-*RB1* alterations in tumors and the impact these alterations have on growth patterns in the eye, metastases and predilection for non-ocular cancers. This study interrogated enucleated retinoblastoma specimens using a MSK-IMPACT clinical next-generation sequencing panel with the aim to correlate them with clinicopathologic characteristics. We found that vitreous seeding (the main reason for eye removal) correlates with copy number variations, specifically 1q gains and 16q loss. We also found that somatic *BCOR* mutations correlate with propensity for metastasis and this offers a molecular pathway for monitoring high risk tumors. In addition, the finding that 11% of these retinoblastoma patients have additional germline mutations (on other chromosomes) that predispose them to a different host of cancers throughout their lives enables more targeted and specific screening strategies.

**Abstract:**

This investigation uses hybridization capture-based next-generation sequencing to deepen our understanding of genetics that underlie retinoblastoma. Eighty-three enucleated retinoblastoma specimens were evaluated using a MSK-IMPACT clinical next-generation sequencing panel to evaluate both somatic and germline alterations. Somatic copy number variations (CNVs) were also identified. Genetic profiles were correlated to clinicopathologic characteristics. *RB1* inactivation was found in 79 (97.5%) patients. All specimens had additional molecular alterations. The most common non-*RB1* gene alteration was *BCOR* in 19 (22.9%). Five (11.0%) had pathogenic germline mutations in other non-*RB1* cancer predisposition genes. Significant clinicopathologic correlations included: vitreous seeds associated with 1q gains and 16q loss of heterozygosity (BH-corrected *p*-value = 0.008, 0.004; OR = 12.6, 26.7, respectively). *BCOR* mutations were associated with poor prognosis, specifically metastases-free survival (MFS) (nominal *p*-value 0.03). Furthermore, retinoblastoma patients can have non-*RB1* germline mutations in other cancer-associated genes. No two specimens had the identical genetic profile, emphasizing the individuality of tumors with the same clinical diagnosis.

## 1. Introduction

Retinoblastoma is the most common primary pediatric intraocular tumor [1]. More than 100 years ago, a common genetic basis for many retinoblastomas was uncovered. While the pattern of inheritance was found to be a classical autosomal dominant pattern, it was not until the 1970s that it was recognized to be caused by a loss of function of a gene, specifically *RB1*. *RB1* was the first tumor suppressor gene to be cloned (in 1986). Since then, the knowledge of *RB1* alterations has been used as a part of genetic counseling for patients and families, with commercial germline *RB1* testing now available worldwide. Tumors are initiated by biallelic loss of the *RB1* gene, with 90% penetrance [2]. Patients can present with bilateral disease derived from germline *RB1* mutations or with unilateral disease, which commonly have somatic mutations but can be germline in approximately 15% of patients [1].

The importance of the biallelic loss of *RB1* is an example of Knudson’s “two-hit” hypothesis of cancer [3]. *RB1* inactivation is recognized to occur by a variety of mechanisms including single nucleotide variants (SNVs), insertions-deletions (indels), loss of heterozygosity (LOH), large deletions, translocations, promoter hypermethylation and chromothripsis [4,5,6,7,8,9,10]. Unlike other cancers that have one or a small number of hotspot mutations, the suspected hotspots for retinoblastoma account for only 40% of mutations and the remaining are dispersed throughout the gene [11]. Although *MYCN* amplification in association with *RB1* inactivation is known to occur, approximately 2% of retinoblastoma tumors have no derangement in *RB1*, and reportedly exhibit only *MYCN* amplification [12].

Besides tumor initiation by biallelic loss of the *RB1* gene, cytogenetic analysis has identified recurrent copy number variants (CNVs) in retinoblastoma tumors, leading to the proposal of several candidate driver genes other than *RB1* such as *KIF14*, *MYCN*, *DEK*, *E2F3*, *RBL2/p130*, and *NGFR* [4,13,14,15,16]. Recently, whole exome sequencing has identified additional putative drivers in *BCOR* and *CREBBP* [7,8,17]. However, the clinical significance of these additional gene mutations, along with the specific mechanism of *RB1* loss, remains to be fully elucidated. While next-generation sequencing (NGS) has been used to analyze retinoblastoma, most studies are either limited to detecting germline mutations, constrained to only a handful of known mutations to occur in retinoblastoma or without cliniopathological correlations [7,12,17,18,19,20]. This study investigated retinoblastoma tumors with hybridization capture-based NGS, analyzed using a MSK-IMPACT platform [21,22]. This platform analyzes 468 genes for somatic analysis, 88 genes for germline analysis, and can identify DNA sequence variants, copy-number alterations, and select rearrangements. This genetic data was analyzed for correlations with clinicopathological features.

## 2. Results

### 2.1. Genomic Landscape of RB1 Alterations in Retinoblastoma

As expected, pathognomonic *RB1* alterations were identified in 97.5% (79/81) of patients (Figure 1A), of which ten were found in specimens from multiple patients in this cohort (Figure 1B).

Of these, seven are shared exclusively in the somatic context, while three can be found in the germline and soma (c.1072C>T, c.1654C>T and c.1981C>T). Furthermore, of the forty-six somatic *RB1* mutations that could be assessed for clonality, only five were subclonal. Two additional specimens did not have detectable *RB1* alterations in either somatic or germline and no other putative drivers were identified in these specimens. One specimen had two additional subclonal mutations in *BIRC3* (p.A385D), a variant of unknown significance, and *PBRM1* (p.G176*), a truncating mutation. The other specimen did not have any identifiable genomic alterations except for a gain of 6p. The lack of *RB1* alterations in two samples is consistent with either RB1 negative tumorigenesis of retinoblastoma or a mechanism of RB1 inactivation not detectable by the methods used in this paper.

Biallelic loss of RB1 was identified in 79.5% (66/83) of specimens (Figure 2A).

Of the 66 specimens with two hits to *RB1*, ten had somatic homozygous deletion of *RB1*, while one had a germline heterozygous loss of exons 7–11 in RB1 followed by a somatic LOH event leading to two dysfunctional copies of *RB1*. Of the remaining 53, 38 (71.6%) had a mutation followed by an LOH event to lose the wild-type copy of *RB1* while 15 had two mutations in *RB1*. Two specimens (with additional methylation testing) had hypermethylated promotors and LOH over the *RB1* locus. Twelve specimens had apparent heterozygous alterations to RB1 as detected by MSK-IMPACT. This suggests other modes of silencing *RB1* may be occurring in these patients that are not detectable by MSK-IMPACT. Evidence of this possibility is highlighted by two patients for which *RB1* methylation testing was performed. However, not all samples were tested for methylation status, and as such the prevalence of *RB1* silencing via this mechanism in patients with a single genomic alteration is unknown. These results demonstrate the variety of different mechanisms leading to RB1 loss, though none of the mechanisms were associated with any particular clinical outcome.

Of the 42 matched specimens with LOH over the *RB1* locus, 31 (73.8%) were copy-neutral LOH (CN-LOH), where two identical alleles are present, in contrast to LOH via the loss of an allele. In the case of *RB1*, this means that two dysfunctional mutant alleles are present but for the rest of the genes on 13q, two supposedly wild-type alleles can be expressed. Compared to other tumor suppressor genes, our specimens showed much higher CN-LOH at *RB1* (*p*-value < 2.2 × 10^−16^, OR 19.8) (Figure 2B,C). Additionally, when compared to other cancer types, *RB1* is subject to CN-LOH at much higher rates in retinoblastoma than in other cancers that also contain mutations in *RB1* (*p*-value < 2.2 × 10^−16^, OR 44.4) (Figure 2D). This preference for CN-LOH of *RB1* in retinoblastoma may reflect the necessity of a gene or genes on 13q to be expressed from two alleles or there may be genes essential to the fitness of the cell type that would be left vulnerable by the presence of a single allele.

### 2.2. Evidence of Intertumoral Genetic Heterogeneity

Two patients with bilateral disease and germline *RB1* alterations had two specimens collected, one from each eye. As expected, within each patient the germline mutations were consistent across the specimens. However, each specimen had different somatic *RB1* second hits. In the first patient, there was a germline loss in *RB1* found in both eyes but different oncogenic somatic variants in each eye, i.e., splice site alteration p.X566_splice and truncating p.V654Cfs*4. In the second patient, the germline alteration was a splice mutation (*RB1* X775_splice). In one eye, this was followed by a somatic LOH while the other eye had a somatic truncating mutation (*RB1* p.S807*). The samples from these patients show that the germline event is not predictive of a particular somatic event and also demonstrates the importance of considering the specimens separately as they often have different etiologies even in the same patient.

In our cohort, three patients had evidence of more than two potential loss of function alterations in *RB1*. All three patients had a germline mutation, an LOH event leading to the somatic loss of the WT copy of *RB1* and an additional somatic mutation. The presence of the second somatic hit in the RB1 germline background is suggestive of two distinct molecular alterations within the specimen. In two of the patients, the additional somatic mutation is at very low allele frequency (Figure 2E) in comparison to the purity of the sample, consistent with the presence of two genetic alterations within the same tumor sample. Under this scenario, the LOH would also be expected to be heterogeneous, however, it appears clonal in both cases and may represent differences between mutation and CNA methods in terms of sensitivity in detecting subclonal events. In the third individual, the germline mutation is silent but previous work suggests that it may impact splicing [25]. However, both somatic events (LOH and mutation) appear clonal (Figure 2E), indicating they must occur in the same tumor cells. In this case, the role of the germline mutation is less clear and its presence in the patient may be coincidental.

### 2.3. Non-RB1 Alterations in Retinoblastoma

We assessed all alterations and identified genes that were frequently altered in our patient population or were identified in other publications about retinoblastoma [7,8,17] (Figure 1A). Aside from RB1, the most commonly somatically mutated genes among the specimens were BCOR in 19 (22.9%), *RPTOR* in 3 (3.6%), *TERT* in 3 (3.6%), *MSH3* in 3 (3.6%), *ARID1A* in 3 (3.6%), *MYCN* in 3 (3.6%), *TSC2* in 2 (2.4%), and *CREBBP* in 1 (1.2%) (Figure 1A). All *BCOR* mutations detected are putative loss of function mutations and therefore likely oncogenic. *BCOR* has been previously proposed to have an association with worse prognosis [7,17]. Here, *BCOR* mutations were associated with a poor prognosis for metastasis-free survival from date of enucleation (nominal *p*-value = 0.03) (Figure 3A), but not with “higher risk” histopathological features. Besides *BCOR*, the ten most frequent non-*RB1* gene alterations were found three times, each, in our cohort of 83 (3.6%) patients. Among these ten genes, only four (*TERT*, *MSH3*, *ARID1A*, and *HLA-A*) had any mutations that were considered putative drivers [26]. Eleven additional genes have singleton, somatic alterations that were considered putative drivers. Only *BCOR* appeared in high enough frequency in our population to assess association with clinical features.

In contrast, none of the mutations in *FAT1* and *RPTOR* are truncating or fall at any hotspot location, as such they are considered variants of unknown significance (Figure 1A). In addition to mutations, three patients exhibited *MYCN* amplification, in the presence of an *RB1* mutation, which has been shown previously to be prevalent in retinoblastoma [27]. Overall, the mutation burden was generally low in these retinoblastoma samples (0–3 mutations per megabase).

In addition to germline pathogenic variants in *RB1*, five of 42 patients (12.0%) consented for germline MSK-IMPACT testing had pathogenic or likely pathogenic germline mutations in non-*RB1* cancer predisposition genes. Mutations were detected in base excision repair: *NTHL1* c.268C>T (p.Gln90*); nucleotide excision repair: *ERCC3* c.1354C>T (p.Arg452*), mis-match repair: *MSH3* c.1341-1G>T (p.X447_splice) splice site mutation and cell cycle: *CHEK2* c.470T>C (p.Ile157Thr) and transcription factor: *MITF* c.1255G>A (p.Glu419Lys) genes. Notably, two of these patients harbored heterozygous alterations in cancer predisposing genes associated with autosomal recessive cancer predisposition syndromes (*NTHL1* and *MSH3*). There was no evidence of microsatellite instability in the patient with germline *MSH3*.

### 2.4. Copy Number Alterations and Loss of Heterozygosity in Retinoblastoma

Several recurrent arm-level CNAs were identified, including 1q gain (67.0%), 2p gain (32.0%), 6p gain (59.0%), LOH 13q (51.0%), LOH 16q (43.0%) (Figure 3B). Specifically, gains on 1q and loss of heterozygosity on 16q were associated with vitreous seeds (BH-corrected *p*-value = 0.008, 0.004; OR = 12.6, 26.7, respectively) (Figure 3C). Gains on 1q and losses on 16q were not significantly correlated with each other (*p*-value = 0.37) (Figure 3D) suggesting that they do not need to co-occur to present with vitreous seeds. A model of vitreous seed presence including both 1q gains and 16q losses showed significant association suggesting that they both play an independent role in vitreous seeds (model *p*-value = 8.2 × 10^−8^) with both being predictive (1q *p*-value = 1.3 × 10^−4^, 16q *p*-value = 2.3 × 10^−4^). No other CNAs were associated with any other clinical features.

## 3. Discussion

The genetic basis of retinoblastoma was long believed to be predominantly limited to biallelic loss of *RB1*. Other non-*RB1* genes have previously been identified [7,8,17] and their clinical relevance sparingly elucidated. However, the present findings based on 83 enucleation retinoblastoma specimens demonstrate that the genetic landscape is more complex and extends our current knowledge. While we detected mutations, loss of heterozygosity, and homozygous deletions leading to biallelic inactivation of *RB1*, we also identified large-scale CNVs and somatic and germline mutations in other genes not on chromosome 13.

The majority of series investigating next-generation sequencing (NGS) in retinoblastoma have used this technology for germline RB1 detection, and copy number alteration details are not consistently available [18]. Few studies focus on using NGS for tumor specimens, in detecting mutations other than RB1 and no more than a dozen other genes known to occur in retinoblastoma, or in correlating these findings with clinic-histopathologic features [7,12,17,19,20]. One study used NGS of 500 oncogenes with clinical correlation, but in only 30 tumor samples of retinoblastoma [28]. Our study investigated 468 genes in 83 retinoblastoma tumor specimens with cliniopathological correlations.

Previous work has shown recurrent chromosomal abnormalities in retinoblastoma including gains of 1q, 2p, 6p and loss of 13q and 16q [4,13,14,15]. Despite extensive knowledge on the presence of CNVs in retinoblastoma, clinicopathologic correlates are limited to correlations with age, differentiation, pathological features and disease laterality [6,15,27]. Loss of 16q has been associated with diffuse vitreous seeding with *CDH11* being proposed as the candidate gene [5,29]. In the present study, both 1p gain and 16q loss were independently associated with the presence of vitreous seeds. However, in our cohort, the role of *CDH11* is unclear in the absence of mutations. Although 6p was recurrently gained in our cohort, contrary to a previous report, it was not associated with aggressive histopathological features or a poor prognosticator of ocular survival [30,31]. This difference may reflect the presence of enucleated-only specimens. Intriguingly, one unilateral specimen with confirmed wildtype *RB1* did possess 6p gain. Indicating either a deficiency in testing sensitivity or strategy, or alternatively, 6p gain may represent an independent biomarker for retinoblastoma in the presence of wildtype *RB1*.

We found that, compared to other cancers, *RB1* in retinoblastoma is significantly and disproportionately more likely to display copy-neutral LOH rather than heterozygous loss LOH. This unique characteristic is previously unreported in retinoblastoma. Other studies have shown the presence of CN-LOH in different cancers, some of which highlight CN-LOH of 13q [32,33,34,35,36]. In our data, we focused on tumors with mutations in *RB1* but CN-LOH is a mechanism that may affect different tumor suppressor genes in other tumor types. While we do not currently have an explanation for this phenomenon, it may suggest that some gene (or genes) on chromosome 13 are essential for cell survival and are needed at two copies.

An additional clinical correlation is observed in the three specimens with three hits to *RB1.* These may represent multiple clones in the same eye and explain the well-recognized observation that multifocal tumors in the same eye have distinct features.

*BCOR* is located at Xp11.4, encodes the BCL6 corepressor and as such is a transcription regulatory factor. *BCOR* mutations are present in a number of malignancies; and correlates with poor cancer prognosis [37,38]. Furthermore, *BCOR* expression aids in eye development and is highly expressed in the lens and retina [39]; and this latter property may explain why retinoblastomas are prone to mutated *BCOR*. In our cohort, it was most common non-*RB1* gene abnormality, occurring in 23% of specimens (higher than previously published series (13%) [7,17]) and was associated with poor prognosis, specifically worse metastases-free survival, but not with high-risk pathological features. Identification of a *BCOR* mutation, either in an enucleated specimen or through cell free DNA (cfDNA) analysis, may influence surveillance strategies or preventative measures for metastatic retinoblastoma. It could mark minimal residual disease and/or be informative of treatment response through quantitative cfDNA.

Future studies may expand upon these results by considering more downstream analytes, such as DNA methylation or RNA expression, to assess biological differences between patients either dependent or independent of DNA mutations. For example, recent studies have considered genome-wide methylation and found expression patterns unique to retinoblastoma, suggesting their importance to oncogenesis [40,41]. Further clinical assessment of patient phenotypes alongside various biological measurements may yield important insight into the progression of retinoblastoma from DNA mutation to unique clinicopathological features of retinoblastoma.

Five patients had non-*RB1* cancer-associated germline mutations in keeping with other cancer cohorts in which 12–17.5% patient had germline mutations [42]. However, whereas germline cancer-associated mutations are associated with metastatic disease, and therapy-specific prognosis in other cancers [43,44], to date our five patients have demonstrated intraocular disease only. This raises the question whether germline variants influence tumorigenesis, the clinical course or subsequent disease risks (metastatic disease or second cancers).

Of the non-*RB1* gene alterations in our cohort, only *MYCN* on 2p24.3 and *TSC2* on 16p13.3 correspond with common CNVs that characterize retinoblastoma [4,13,14,15,16]. Previous studies have reported mutations in *CREBBP* and we found one example in our patients [7,8,17]. We identified additional variants that are recognized as oncogenic in other cancer types [26]. However, in our cohort, each of these occurs at low frequency, limiting our ability to assess their functionality in retinoblastoma, as none of the genes are commonly associated with retinoblastoma. The remaining gene alterations do not correspond with typical oncogenic variants associated with retinoblastoma or other cancer types. For example, *FAT1* acts as both a tumor suppressor and oncogene and occurs in bladder cancer, head and neck cancers, breast and colorectal cancers. Published literature suggests it increases invasiveness of tumors [45]. *RPTOR*, along with *TSC2*, is a part of the mTOR signaling pathway known to affect many different types of cancer [46]. However, neither *FAT1* nor *RPTOR* have putative drivers in our dataset.

## 4. Materials and Methods

An institutional review board approval was obtained from Memorial Sloan Kettering Cancer Center (#17-049). Eligible patients included those with retinoblastoma who had undergone a primary or secondary enucleation, and those included were enucleated between May 2006 and August 2019. Patients were consented for either somatic or somatic and germline testing via MSK-IMPACT. Prior commercial germline testing for *RB1* was also available for some patients.

Clinicopathologic data was collected for each patient and retinoblastoma tumor. This data included gender, laterality, prior treatment, ocular hypertension, rubeosis, vitreous seeds (including seed class: dust, spheres, or clouds) or subretinal seeds, histopathologic features (invasion into the choroid (massive or not), sclera, ciliary body, anterior chamber, optic nerve pass the lamina cribrosa (yes or no) optic nerve to the cut section), second cancers, occurrence and site of metastatic disease (defined as orbital or other extraocular metastasis). “Higher risk” eyes were defined as those with high risk pathological features defined as massive choroidal invasion, ciliary body/iris invasion or optic nerve invasion past the laminar cribrosa.

### 4.1. Isolation and Purification of DNA

Microdissection was performed on 83 formalin-fixed and paraffin-embedded (FFPE) samples on 10 µm-thick unstained sections, using hematoxylin and eosin-stained (H&E) sections as a guide. The DNeasy Tissue Kit (Qiagen, Hilden, Germany) was used for DNA extraction according to the manufacturer’s recommendations. The Nano-Drop 8000 (Thermo Scientific, Waltham, MA, USA) and Qubit (Life Technologies, Carlsbad, CA, USA) were employed to quantify the extracted DNA. The minimum concentration of formalin fixed paraffin embedded DNA was 250 ng.

### 4.2. Exon-Capture Sequencing

Targeted next-generation sequencing via MSK-IMPACT was performed on DNA extracted from tumor and matched normal DNA samples, with somatic mutations (substitutions and small insertions and deletions), gene-level focal copy number alterations, and structural rearrangements detected [21,22]. Germline variants were evaluated in 88 cancer predisposition genes in 42 patients [47].

### 4.3. Outside Testing of RB1 Alterations

Outside commercial testing was performed by multiple companies as routine clinical care on 23 specimens. Each company performed their individual assay, which identifies mutations, methylation, and copy number alterations at the *RB1* locus. In this work, we used methylation status and germline *RB1* mutations for patients not consented to germline testing via MSK-IMPACT. In addition, germline CNVs in *RB1* and somatic CNAs from outside testing are reported for cases sequenced via MSK-IMPACT unmatched (without a normal tissue comparator from the patient). MSK-IMPACT identified all *RB1* mutations found by outside testing in specimens that had both outside germline testing and germline testing via MSK-IMPACT.

### 4.4. Biostatistics

Total, allele-specific and integer copy number as well as tumor purity and ploidy were identified in the 74 specimens with matched normal tissue using FACETS (Fraction and Allele-Specific Copy Number Estimates from Tumor Sequencing) v0.5.14 [48] and the clonality of mutations was calculated as described [49]. For the nine cases sequenced without matching normal tissue, copy number alterations were identified using the GATK DepthOfCoverage tool as previously described [49] Microsatellite instability was estimated using MSIsensor [50]. Kaplan–Meier estimates of event-free survival were calculated starting from the date of enucleation to the detection of metastasis, and correlated to genetic alterations. Four individuals were removed from the metastasis-free survival estimates because they presented with metastasis at, or before, the time of enucleation. When multiple tests were performed, Benjamini–Hochberg (BH) multiple testing correction was used [51].

For the purposes of this manuscript, LOH is the state of total copy number 1 and lower copy number 0 (1,0). Copy-neutral LOH (CN-LOH) is 2,0, while reciprocal LOH (RLOH) [52] is total copy number greater than two, while lower copy number is 0. Tumor suppressor genes were defined from OncoKB [26]. LOH rates at the *RB1* locus across the MSK cohort (frozen in Jan, 2019 with 39,846 tumor specimens) used default FACETS fits (not manually reviewed) that passed QC.

## 5. Conclusions

In conclusion, there is true genetic diversity among retinoblastoma tumors. Based on an analysis of 468 genes, no two specimens analyzed had identical pattern of mutations (or type of *RB1* loss), LOH or CNV, emphasizing the individuality of tumors with the same clinical diagnosis. We identified clinicopathologic correlations with genetic aberrations including the association of vitreous seeds with 1q gains and 16q loss. Our discovery of *BCOR* mutations being correlated with the development of metastatic disease may guide future systemic surveillance or management of these patients. Insights gained on small numbers of specimens include the observation that a single eye with multifocal tumors may include more than two identifiable forms of *RB1* loss, suggesting intertumoral heterogeneity, and that in wildtype *RB1*, 6p gain may be sufficient for retinoblastoma tumorigenesis. Finally, retinoblastoma patients can have non-*RB1* germline mutations in other cancer-associated genes. In the future, we hope to discover how these alternate non-*RB1* germline mutations may influence the primary ocular retinoblastoma, the metastatic potential or additional primary malignancies.

## Figures and Tables

**Figure 1 cancers-13-00149-f001:**
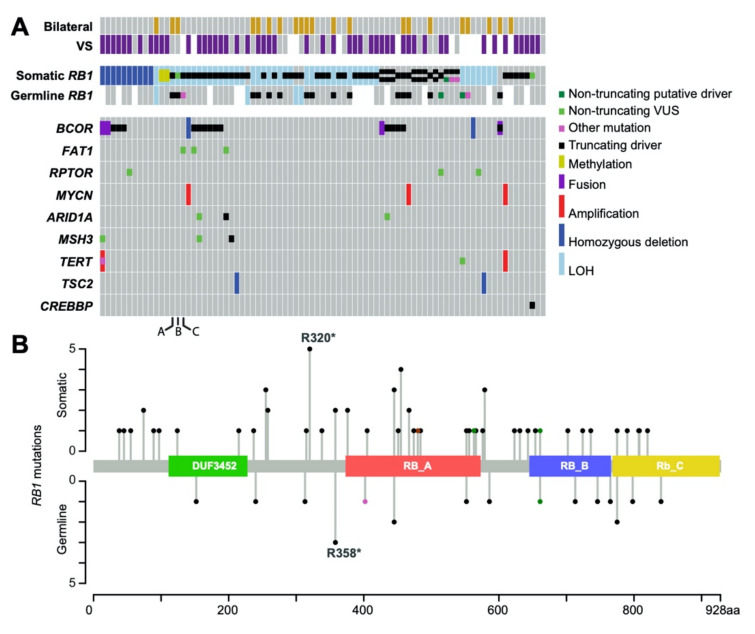
Mutational landscape of retinoblastoma. (**A**) Oncoprint of genetic, clinical, histopathologic features of 83 retinoblastoma eyes. Top section depicts clinicopathologic features of the specimens including presence of vitreous seeds (VS) and laterality. Because of the importance of RB1, we annotated germline and somatic RB1 hits separately and also included loss of heterozygosity and methylation status, which we display when multiple mutations occur. The rest of the genes are annotated, combining germline, somatic and multiple alterations. Missing data is white. A–C annotate three samples with three hits to RB1. (**B**) Plot of the somatic (top) and germline (bottom) alterations of RB1, known RB1 domains marked along the axis. The height of each line indicates the number of alterations that occur at that position within the protein, with the most frequent mutation labeled in both somatic and germline.

**Figure 2 cancers-13-00149-f002:**
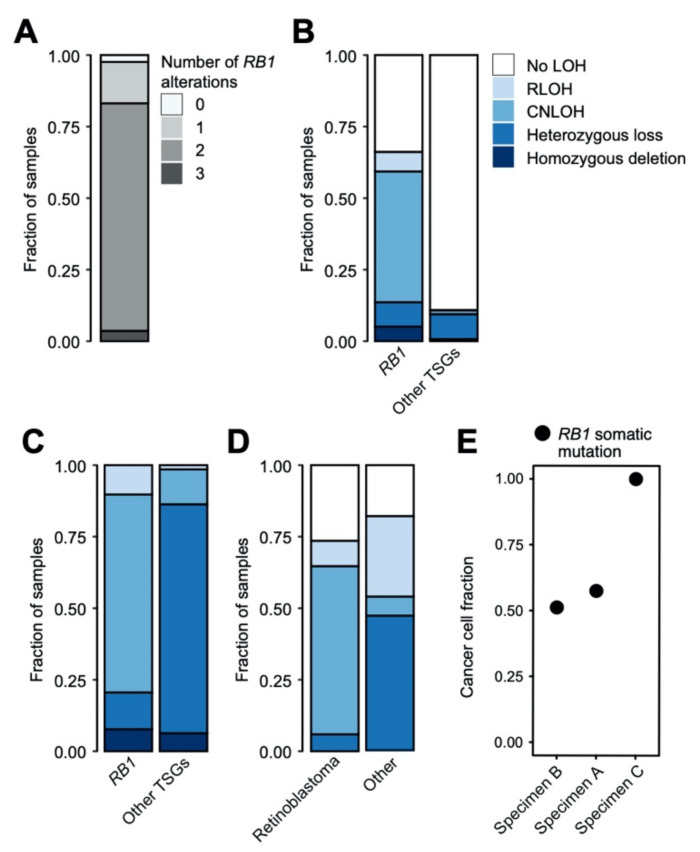
Mechanisms of inactivation of RB1. (**A**) Stacked bar graph depicting number of hits to RB1 in 83 eyes. (**B**,**C**) Bar plot showing the types of loss of heterozygosity that occurs in retinoblastoma samples at the RB1 locus vs. loci of other tumor suppressor genes as annotated by OncoKb [23,24]. (**C**) Bar plot of the proportion of each type of loss of heterozygosity across the specimens, excluding those without any type of loss. There is a significantly higher proportion of CN-LOH at the RB1 locus (*p* < 2.2 × 10^−16^, OR = 19.8). (**D**) Bar plots comparing the types of loss of heterozygosity at the RB1 locus in malignancies with RB1 mutations. Retinoblastoma samples are compared to all other types of malignancies. There is significantly higher proportion of CN-LOH at the RB1 locus in retinoblastoma compared to other malignancies (*p* < 2.2 × 10^−16^, OR = 44.4). (**E**) For the somatic variants in the specimens with three detectable hits to RB1, we calculated the variant allele frequency corrected for purity of the sample and ploidy of the region (cancer cell fraction). Each column is the variant from a different specimen. Specimens are labeled based on Figure 1A.

**Figure 3 cancers-13-00149-f003:**
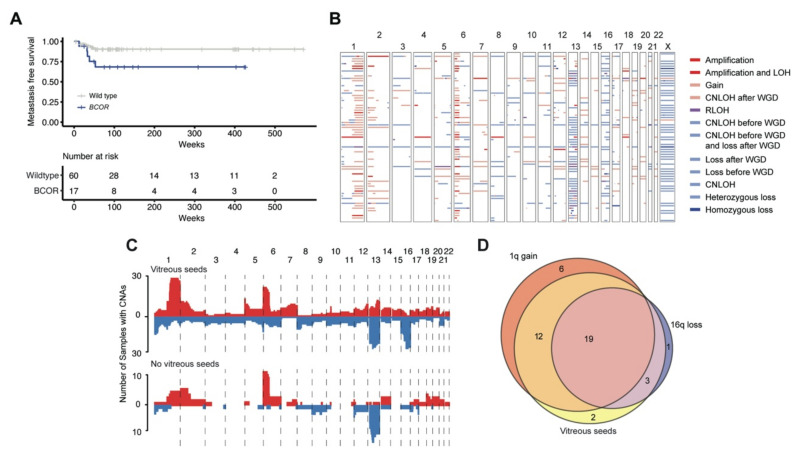
Correlation of genomic features with clinical characteristics. (**A**) Kaplan–Meier graph comparing metastases-free survival between specimens with a BCOR alteration and those without. BCOR is nominally correlated with metastases-free survival (*p*-value = 0.03). (**B**) Copy number alterations of 69 eyes. Each row is an individual and each column is a different chromosome. White space shows where no gains or losses occur. Losses are shown in shades of blue while gains are shown in shades of red. (**C**) Plot showing the genomic copy number changes in individuals with vitreous seeds vs. without. The genome was binned into 100,000 bp regions and the height of each region shows how many specimens had a gain or loss in a given bin. Any gain is shown in red on the top of each plot while any sort of loss of heterozygosity is shown in blue at the bottom of each plot. There is a significant enrichment for 1q gain and 16q loss in individuals with vitreous seeds (BH-corrected *p*-value = 0.008 and 0.004, respectively). (**D**) Venn diagram of specimens that contain 1q gain, 16q loss and vitreous seeds. Limited to 43 samples that had annotation of vitreous seeds and FACETS fits to call alterations on 1q and 16q.

## Data Availability

IMPACT mutation calls were deposited via the cBioPortal for Cancer Genomics (https://cbioportal.mskcc.org/study/summary?id=rbl_mskcc_2020).

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
