# Peer review of "Molecular Changes in Retinoblastoma beyond RB1: Findings from Next-Generation Sequencing"

_cancers, 2021, doi:10.3390/cancers13010149_

Round 1
Reviewer 1 Report
Francis and colleagues examine 83 retinoblastomas using a clinical next-generation sequencing panel and correlate the molecular alterations found with clinic-pathologic characteristics. They report that vitreous seeding (the main reason for eye removal) correlates with 1q gain and 16q loss. They also found that somatic BCOR mutations correlate with propensity for metastasis. Aside from RB1 the most commonly somatically mutated genes among the specimens were BCOR in 19 (22.9%), FAT1 in 5 (6.0%), MAP3K1 in 4 (4.8%), TSC2 in 3 (3.6%), TERT in 3 (3.6%), RPTOR in 3 (3.6%), PTCH1 in 2 (2.4%) and CREBBP in 1 (1.2%). Finally, they report that 11% of these retinoblastoma patients have additional germline mutations that predispose them to other types of cancer.
Overall, this is a straightforward descriptive study, which does not uncover many novel molecular findings, as most have been demonstrated in prior reports. However, relatively few prior high-quality publications exist integrating copy number and broader mutational profiles with clinical and histopathological factors in retinoblastoma. This work will therefore be of significant interest to those in the field, particularly as the number of cases examined is relatively large. However, it could be significantly improved in a number of ways.
One major issue is the lack of a Table providing basic clinical and demographic data for each case. In either the same table, or other ones, specific molecular alterations should be shown. Without tabular presentation of the data, it will not be possible for others to fully analyze or build on this work.
It is interesting that poor clinical outcome was reportedly associated with BCOR alterations. It would be helpful if more details could be presented on what association(s), if any, were seen in BCOR altered cases with other discrete high risk features such as massive choroidal invasion, optic nerve invasion, etc.
Additional issues:
- They showed various genes are mutated such as BCOR, FAT1, MAP3K1, TSC2, TERT, RPTOR, PTCH1 in Figure 1A. Did any of the other alterations show a trend towards association with outcome overall, or specific high risk factors.
- While vitreous seeding has historically been a major reason for eye removal, increased use of intravireal chemotherapy has reduced morbidity in these cases, reducing the impact of this association with BCOR to a degree.
- During what period of time were these cases accrued? Was treatment similar in cases for which clinical outcome is presented?
- While the authors cite specific genetic alterations in retinoblastoma which have been previously reported, a sentence or two in the Introduction or Discussion placing this NGS analysis more broadly in the context of prior work (ie how many cases were examined in previous studies using NGS or copy number analyses) would be helpful.
In summary, this is a compact descriptive study which does not break significant new conceptual ground. However, it does have potentially great value as a fairly large series of retinoblastoma cases with detailed NGS analysis including copy number profiling. To maximize this impact on the field, it will be important for the investigators to provide tabular summaries of the demographic, clinical and molecular data (perhaps in supplemental figures).
Author Response
Thank you and the reviewers for your comments regarding our manuscript ID cancers-1007335, “Molecular Changes in Retinoblastoma Beyond RB1: Findings from Next-Generation Sequencing”. We have found these comments useful in improving our manuscript and offer the following itemized responses:
Comments and Suggestions for Authors
Reviewer 1: One major issue is the lack of a Table providing basic clinical and demographic data for each case. In either the same table, or other ones, specific molecular alterations should be shown. Without tabular presentation of the data, it will not be possible for others to fully analyze or build on this work.
The authors agree that a table collating a summary of demographic information and genetic information would enhance the paper and potentially act as a resource for others to expand on this work. We have included this table as a cBioportal instance and have provided the uniform resource locator (URL) in the materials and methods on line 536, “IMPACT mutation calls were deposited via the cBioPortal for Cancer Genomics (https://cbioportal.mskcc.org/study/summary?id=rbl_mskcc_2020). This instance will be made public if, and when the paper is accepted for publication.
- It is interesting that poor clinical outcome was reportedly associated with BCOR alterations. It would be helpful if more details could be presented on what association(s), if any, were seen in BCOR altered cases with other discrete high risk features such as massive choroidal invasion, optic nerve invasion, etc.
The reviewer raises an important point regarding the association of BCOR alterations with high-risk features. In the original manuscript, on line 217, we state, “Here, BCOR mutations were associated with a poor prognosis for metastasis-free survival (nominal p-value 0.02) (Figure 3A); but not with “higher risk” histopathological features.” With higher risk features being defined on line 317 as, “as those with high risk pathological features defined as massive choroidal invasion, ciliary body/iris invasion or optic nerve invasion past the laminar cribrosa.”
Reviewer 1: They showed various genes are mutated such as BCOR, FAT1, MAP3K1, TSC2, TERT, RPTOR, PTCH1 in Figure 1A. Did any of the other alterations show a trend towards association with outcome overall, or specific high risk factors.
The reviewer brings up another important point regarding the association of other non-RB1 alterations with high-risk features. After submission, we updated our mutation filtering and have modified the oncoprint accordingly. Unfortunately, after filtering we do not find any gene mutated in more than 3 patients each, other than RB1 and BCOR. This low frequency inhibits us from performing statistical tests of association with clinical features. Furthermore, only RB1 and BCOR have multiple alterations that are predicted to have an oncogenic effect. Variants of unknown significance (VUS) are difficult to interpret because there is not a clear suggestion of how they affect the protein structures or the function of the protein. For these two reasons, we did not perform associations with high-risk factors for the other genes in the oncoprint. We have clarified this point in the following statement on line 188: Besides BCOR, the ten most frequent non-RB1 gene alterations were found three times, each, in our cohort of 83 (3.6%) patients. Among these ten genes, only four (TERT, MSH3, ARID1A, and HLA-A) had any mutations that were considered putative drivers [49]. Eleven additional genes have singleton, somatic alterations that were considered putative drivers. Only BCOR appeared in high enough frequency in our population to assess association with clinical features.
Reviewer 1: While vitreous seeding has historically been a major reason for eye removal, increased use of intravireal chemotherapy has reduced morbidity in these cases, reducing the impact of this association with BCOR to a degree.
During what period of time were these cases accrued? Was treatment similar in cases for which clinical outcome is presented?
The reviewer asks a pertinent question regarding the influence of treatment on the potential prognostic value of non-RB1 mutations such as BCOR. These eyes included in the present study were enucleated between May 2006 and August 2019, and for all eyes in this cohort, treatment consisted of either primary or secondary enucleation. Even though are analysis is limited to enucleated eyes only, we hope that our results with be applicable to genetic information that can be obtained from non-invasive methods (such as cell free DNA analysis) and therefore, in the context of eye-conserving treatment (such as intravitreous chemotherapy). To address the reviewer’s questions, we have added the following statements on line 264, “Eligible patients included those with retinoblastoma who had undergone a primary or secondary enucleation, and those included were enucleated between May 2006 and August 2019.”
Reviewer 1: In our cohort, BCOR appears to be related to metastasis-free survival but not clinical or histopathological features.
While the authors cite specific genetic alterations in retinoblastoma which have been previously reported, a sentence or two in the Introduction or Discussion placing this NGS analysis more broadly in the context of prior work (ie how many cases were examined in previous studies using NGS or copy number analyses) would be helpful.
We agree that including more contextual information regarding the present literature on NGS for retinoblastoma would be worthwhile.
In the introduction, on line 75, the following sentence has been added, “While next-generation sequencing (NGS) has been used to analyze retinoblastoma, most studies are either constrained to detecting germline mutations or limited to only a handful of known mutations to occur in retinoblastoma.”
And in the discussion, the following has been added to line 227. However, if the reviewers feel this is repetitive, we would be open to removing this section.
“The majority of series investigating Next-Generation Sequencing (NGS) in retinoblastoma have used this technology for germline RB1 detection and copy number alteration details are not consistently available. Few studies focus on using NGS for tumor specimens, in detecting mutations other than RB1 and no more than a dozen other genes known to occur in retinoblastoma, or in correlating these findings with clinic-histopathologic features. One study used NGS of 500 oncogenes with clinical correlation, but only in 30 tumor samples of retinoblastoma. Our study investigates 468 genes in 83 retinoblastoma tumor specimens with cliniopathological correlations.”
Reviewer 2 Report
Francis, JH et al. report the use of hybridization capture-based NGS to identify mutations in 468 genes for somatic analysis, 88 genes for germline analysis, can identify DNA sequence variance, CNV, and select chromosomal abnormalities in 83 enucleated retinoblastoma specimens with the purpose of correlating these genetic changes to clinicopathological features. Retinoblastoma is a rare form of pediatric eye cancer and mutations in the RB1 gene are known to be responsible for most of these cancers. RB1 was the first cloned tumor suppressor and has been the target of cancer research for more than 30 years. The authors show a variety of genetic changes that lead to the loss of RB function including mutations (germline and somatic), DNA methylation of the RB1 promoter, LOH. Among the findings of interest, the authors show that in retinoblastomas there was a high level of copy-neutral LOH (when two identical alleles are present), which is not the case for other cancers, eyes from the same patients can have different genetic alterations, some specimens showed up to three different alterations in the RB1 gene, and that there was increased incidence of chromosome 1q gain and chromosome 16q loss. The article is clearly written and of interest for the scientific community that works on RB. Below are a few comments that can help improve the manuscript before publication.
Comments:
- 2: E panel is missing labels on the X axis. I am not sure if this was done on purpose due to the long labels (as described in the legend) but I think it would be easier to the readers if the figure had labels.
- I think the article would be improved by a short discussion or mention, along with references regarding the function of the other potential tumor suppressor genes that were identified in the study. For instance, the authors identify mutations in BCOR and say that this gene is highly expressed in the lens and retina, but do not really mention the gene function. Additional details regarding gene function for these other TS genes would be helpful.
Author Response
Thank you and the reviewers for your comments regarding our manuscript ID cancers-1007335, “Molecular Changes in Retinoblastoma Beyond RB1: Findings from Next-Generation Sequencing”. We have found these comments useful in improving our manuscript and offer the following itemized responses:
Reviewer 2: 2: E panel is missing labels on the X axis. I am not sure if this was done on purpose due to the long labels (as described in the legend) but I think it would be easier to the readers if the figure had labels.
We thank the reviewer for making this observation. We have revised panel 2E so that it has labels.
Reviewer 2: I think the article would be improved by a short discussion or mention, along with references regarding the function of the other potential tumor suppressor genes that were identified in the study. For instance, the authors identify mutations in BCOR and say that this gene is highly expressed in the lens and retina, but do not really mention the gene function. Additional details regarding gene function for these other TS genes would be helpful.
The authors thank the reviewer for the opportunity to address these important questions, and we agree that additional details regarding gene function would enhance the discussion. We have revised the manuscript first by revising line 239, so it now reads, “BCOR is located at Xp11.4, encodes the BCL6 corepressor and as such is a transcription regulatory factor. BCOR mutations are present in a number of malignancies; and correlates with poor cancer prognosis [27] [28]. Furthermore, BCOR expression aids in eye development and is highly expressed in the lens and retina [29]; and this latter property may explain why retinoblastomas are prone to mutated BCOR.”
In addition, we have added the following paragraph to the discussion:
“Of the non-RB1 gene alterations in our cohort, only MYCN on 2p24.3 and TSC2 on 16p13.3 correspond with common CNVs that characterize retinoblastoma. Previous studies have reported mutations in CREBBP and we find one example in our patients. We identify additional variants that are recognized as oncogenic in other cancer types. However, in our cohort each of these occurs at low frequency limiting our ability to assess their functionality in retinoblastoma, as none of the genes are commonly associated with retinoblastoma. The remaining gene alterations do not correspond with typical oncogenic variants associated with retinoblastoma or other cancer types. For example, FAT1 acts as both a tumor suppressor and oncogene and occurs in bladder cancer, head and neck cancers, breast and colorectal cancers. Published literature suggests it increases invasiveness of tumors [34]. RPTOR, along with TSC2, is a part of the mTOR signaling pathway known to affect many different types of cancer. However, neither FAT1 nor RPTOR have putative drivers in our dataset.
Future studies may expand upon these results by considering more downstream analytes, such as DNA methylation or RNA expression, to assess biological differences between patients either dependent or independent of DNA mutations. For example, recent studies have considered genome-wide methylation and found expression patterns unique to retinoblastoma suggesting their importance to oncogenesis. Further clinical assessment of patient phenotypes alongside various biological measurements may yield important insight into the progression of retinoblastoma from DNA mutation to unique clinicopathological features of retinoblastoma.”